# Risk of cancer with angiotensin-receptor blockers increases with increasing cumulative exposure: Meta-regression analysis of randomized trials

Ilke Sipahi ⓘ *

Department of Cardiology, Acibadem University Medical School, Istanbul, Turkey

* ilkesipahi@gmail.com

## Abstract

Angiotensin-receptor blockers (ARBs) are a class of drugs approved for the treatment of several common conditions, such as hypertension and heart failure. Recently, regulatory agencies have started to identify possibly carcinogenic nitrosamines and azido compounds in a multitude of formulations of several ARBs, resulting in progressive recalls. Furthermore, data from several randomized controlled trials suggested that there is also a clinically increased risk of cancer and specifically lung cancer with ARBs; whereas other trials suggested no increased risk. The purpose of this analysis was to provide additional insight into the ARB-cancer link by examining whether there is a relationship between degree of cumulative exposure to ARBs and risk of cancer in randomized trials. Trial-level data from ARB Trialists Collaboration including 15 randomized controlled trials was extracted and entered into meta-regression analyses. The two co-primary outcomes were the relationship between cumulative exposure to ARBs and risk of all cancers combined and the relationship between cumulative exposure and risk of lung cancer. A total of 74,021 patients were randomized to an ARB resulting in a total cumulative exposure of 172,389 person-years of exposure to daily high dose (or equivalent). 61,197 patients were randomized to control. There was a highly significant correlation between the degree of cumulative exposure to ARBs and risk of all cancers combined (slope = 0.07 [95% CI 0.03 to 0.11], p<0.001), and also lung cancer (slope = 0.16 [95% CI 0.05 to 0.27], p = 0.003). Accordingly, in trials where the cumulative exposure was greater than 3 years of exposure to daily high dose, there was a statistically significant increase in risk of all cancers combined ($I^2$ = 31.4%, RR 1.11 [95% CI 1.03 to 1.19], p = 0.006). There was a statistically significant increase in risk of lung cancers in trials where the cumulative exposure was greater than 2.5 years ($I^2$ = 0%, RR 1.21 [95% CI 1.02 to 1.44], p = 0.03). In trials with lower cumulative exposure to ARBs, there was no increased risk of all cancers combined or lung cancer. Cumulative exposure-risk relationship with ARBs was independent of background angiotensin-converting enzyme inhibitor treatment or the type of control (i.e. placebo or non-placebo control). Since this is a trial-level analysis. the effects of patient characteristics such as age and smoking status could not be examined due to lack of patient-level data. In conclusion, this analysis, for the first time, reveals that risk of cancer with ARBs (and specifically lung cancer) increases with increasing cumulative

**Data Availability Statement:** The data are held in a public repository: https://doi.org/10.5061/dryad. bg79cnpc9. Also all relevant data are actually

within the paper and its Supporting information files.

**Funding:** The author received no specific funding for this work.

**Competing interests:** I have read the journal's policy and the authors of this manuscript have the following competing interests: Dr. Sipahi has received lecture honoraria from Novartis, Boehringer-Ingelheim, Sanofi, Sandoz, Bristol-Myers Squibb, Bayer, Pfizer, Ranbaxy, Servier and ARIS and served on advisory board for Novartis, Sanofi, Servier, Bristol-Myers Squibb, Pfizer, Bayer and I.E. Ulagay. This does not alter my adherence to PLOS ONE policies on sharing data and materials.

exposure to these drugs. The excess risk of cancer with long-term ARB use has public health implications.

## Introduction

Angiotensin-receptor blockers (ARBs) are a widely used class of drugs approved for the treatment of several highly prevalent conditions, including hypertension, heart failure and diabetic nephropathy, as well as for primary prevention of cardiovascular events [1–4]. In 2011, it was estimated that globally over 200 million patients are chronically on ARBs [5]. In 2018, regulatory agencies identified N-nitrosodimethylamine (NDMA), a possible human carcinogen nitrosamine compound in several formulations of a commonly used ARB (valsartan), resulting in a major progressive recall [6, 7]. Subsequently, the Food and Drug administration (FDA) announced that they have started testing all the other drugs in the ARB class for nitrosamines, since the synthesis of other ARBs can have similarities to the synthesis of valsartan and nitrosamines can be a common impurity developing during synthesis of all ARBs [8]. Later on, N-nitrosodiethylamine (NDEA), another possibly carcinogenic nitrosamine, was identified in at least 3 different ARB containing drug products, namely valsartan, losartan and irbesartan, resulting in further recalls in 2018 and 2019 [9–11]. This was followed by the discovery of a third nitrosamine in several ARB drug products, again resulting in recalls [12]. Moreover, throughout 2021, multiple lots of three different ARBs were progressively recalled again, this time due to another potentially carcinogenic impurity, namely azido compounds [13].

Back in 2010, along with my co-investigators, I had reported a comprehensive meta-analysis of long-term large-scale randomized controlled trials and suggested that ARBs can increase the risk of cancer, and specifically lung cancer [14]. Soon after this publication, the regulatory agencies started to run investigations about this risk. In 2011, these agencies concluded that there is no increased risk of cancer with ARBs [15, 16]. Additionally, a multitude of other analyses were subsequently published examining the same issue [17–26]. The results of these analyses were highly heterogeneous, some suggesting no excess risk [21, 24, 26] and others suggesting an increased cancer risk with ARBs [22, 23]. The reasons behind these contradictory conclusions warrant systematic examination, especially in light of the recent progressive recall of several ARB containing drug products due to potentially carcinogenic impurity.

Cumulative exposure is a fundamental factor in the epidemiology of chronic disease, and especially in cancer epidemiology. Unfortunately, the relationship between cumulative exposure to ARBs and risk of cancer has not been examined in the investigations of the regulatory agencies [15, 16] or in other analyses of randomized trials [24–26]. Thus, the objectives of the current analysis were to provide greater insight into the ARB-cancer link by examining the exposure-risk relationship using data from randomized controlled trials and to explore whether different levels of cumulative exposure to ARBs explain the heterogeneity observed in the randomized trials.

## Methods

### Data sources

The aim of this analysis was to utilize the largest and the most reliable dataset available for randomized controlled trials of ARBs. The FDA had performed a trial level meta-analysis, including 155,816 patients from 31 randomized trials [15]. Nevertheless, the FDA did not publish the

details of their methods or results, such as the rates of cancers in the exposed and unexposed patients in each trial or the degree of cumulative exposure for these trials. In November 2013, a public meeting on safety meta-analyses was held by the FDA required by Prescription Drug User Fee Act. During this meeting it was requested from the FDA that the details of all safety meta-analyses should be released, starting with the ARB-cancer meta-analysis. In early December 2013, I sent a letter to the FDA supporting this request and personally asked for the details of their ARB-cancer meta-analysis. However, to the best of my knowledge, patient-level or even trial-level data from that analysis has never been released. On the other hand, the largest publicly available detailed data comes from the ARB Trialists Collaboration, which was a project dedicated to examine the risk of new cancers with ARBs. The sponsors of the randomized trials performed with these drugs provided the data to the ARB Trialists Collaboration [25]. This collaboration ultimately used comprehensive patient-level data, including a total of 138,769 patients from 15 randomized controlled trials [27–41] and concluded that there is no excess risk of new cancers with ARBs. It is highly likely that the ARB Trialists Collaboration data greatly overlaps with the data that the industry provided to the FDA for its official investigation. Therefore, given its size, inclusion of data directly provided by the pharmaceutical companies not available elsewhere and detailed public disclosure enabling examination of cumulative exposure-cancer risk relationship, trial-level data from the published report of the ARB Trialists Collaboration was used for the current analysis [25]. In instances where specific relevant data was not available in the collaboration's report (such as level of study drug compliance), data from the published articles of particular trials or data available at the FDA's website that was posted during the approval of ARBs were used.

Cancer cases, reported as serious adverse events, were systematically collected during all 15 trials [25]. Cancer was a prespecified endpoint of special interest in the LIFE, ONTARGET and TRANSCEND trials and information on the occurrence of malignancies was collected prospectively and in more detail than usual for trials of cardiovascular outcome [27, 28, 41]. Importantly, patients with preexisting cancers before randomization were not included in the analyses of the ONTARGET, TRANSCEND, CHARM, and VALIANT trials [25].

## Data extraction

Data extraction was performed from the report of the ARB Trialists Collaboration and was verified three times [25]. Number of cancers, specific organ cancers (lung, prostate and breast cancers) and total number of patients in the arms of the included trials were extracted. Cancer data stratified according to background angiotensin-converting enzyme (ACE) inhibitor treatment was also available and extracted. To quantify cumulative exposure to ARBs, the mean (or median) duration of follow-up and rates of compliance to study medications were extracted from the same report. The mean (or median) drug doses received by the patients for each trial were extracted from the published reports of all 15 trials [27–41].

## Calculation of cumulative exposure

Cumulative exposure is the product of intensity and duration of exposure to an agent. Regarding the intensity (dose) of exposure, all ARBs included in this analysis had similar dose ranges for the treatment of hypertension approved by the FDA, with the higher daily dose always being fourfold of lower daily dose (i.e. valsartan 80 to 320 mg/day, candesartan 8 to 32 mg/day, losartan 25 to 100 mg/day, telmisartan 20 to 80 mg/day and irbesartan 75 to 300 mg/day). To calculate the average dose that the patients were exposed to in a uniform manner for different ARBs, the mean (or median) daily dose received by the patients in each trial was divided by the daily high dose for that particular ARB. To calculate duration of exposure, the mean (or

median) follow-up duration (in years) was multiplied by % compliance. Thus, cumulative exposure (in years of exposure to daily high dose) was calculated as:

$$Cumulative\ exposure = \frac{mean\ (or\ median)\ daily\ dose\ received}{recommended\ daily\ high\ dose} X\ mean\ (or\ median)\ duration\ of\ follow\ up\ X\ \%\ compliance$$

### Data synthesis

To address the issue of publication bias, funnel plots were generated for all cancers combined, and also specifically for lung cancer. Statistical heterogeneity across the trials was assessed with $I^2$ values. To calculate meta-analytic risk ratios, both fixed-effect and random effects models were used for complete presentation of findings. Inverse variance weighting scheme was used for both types of models. The two co-primary outcomes of this study were 1) the relationship between cumulative exposure to ARBs and risk of all cancers combined and 2) the relationship between cumulative exposure to ARBs and risk lung cancer, since lung cancer was the only specific solid organ cancer whose risk was increased with ARB treatment in our original meta-analysis [14]. The relationships between cumulative exposure and risk of cancers were examined using meta-regression analysis. For completeness, meta-regression was performed with 3 different methods, including fixed effect regression, mixed effects regression-method of moments and the mixed effects regression-unrestricted maximum likelihood method. Meta-regression analyses were also performed in the subgroups of patients receiving background ACE-inhibitor treatment in both treatment arms and in those not receiving such treatment in either arm. Meta-regression analyses were also performed separately for placebo controlled and non-placebo controlled trials. Sensitivity analysis with the one-study out method was also used to examine whether results of the meta-regression analyses were driven by a single trial. Finally, number needed to harm for one excess cancer was calculated using background cancer incidence rates for the mean age of patients in the included trials, as recommended for these types of analyses [42, 43].

Two-tailed p-values less than 0.05 were considered statistically significant. Data were analyzed with Comprehensive Meta Analysis Version 2.2.048 (Biostat Inc, Englewood, New Jersey, USA).

There were no sponsors for this study. The author (IS) had full access to all the data in the study and was responsible for submission of the manuscript for publication.

## Results

### Study characteristics

The main study characteristics of the 15 randomized trials included in the current analysis (as well as the ARB Trialists Collaboration) are presented in Table 1. The original ARB Trialists collaboration included randomized trials enrolling at least 500 patients with an average follow-up of at least 1 year. All trials randomized patients to an ARB or control in a 1:1 fashion except the ONTARGET and VALIANT trials, where patients were randomized to an ARB or an ACE-inhibitor or to combined ARB and ACE-inhibitor in 1:1:1 fashion. In these two trials, patients randomized to ACE-inhibitor only received additional ARB placebo, and those randomized to ARB only received additional ACE-inhibitor placebo. Eleven other 1:1 randomization trials were also placebo controlled. Two of the 1:1 randomization trials had non-placebo (i.e. active) control. Ultimately, a total of 74,021 patients were randomized to an ARB resulting in a total cumulative exposure of 172,389 person-years (of exposure to daily high dose or equivalent). The most commonly used ARBs as the study drug were telmisartan (n = 28,787,

**Table 1. Randomized controlled trials included in the analysis.**

| Study Name | Total n | Condition Studied | ARB used (dose received [a]) | Recommended daily high dose of the ARB studied [b] | Control Drug (dose received [a]) | Duration of follow-up (months) | Adherence to Study Drugs (%) | Average Cumulative Exposure to ARB (years of daily high dose) |
|---|---|---|---|---|---|---|---|---|
| ONTARGET | 25,620 | Cardiovascular disease or diabetes with end-organ damage | Telmisartan (80 mg) or Telmisartan (80 mg) + Ramipril (10 mg) | 80 mg | Ramipril (10 mg) | 56 | 80 | 3.73 |
| TRANSCEND | 5,926 | Angiotensin- converting enzyme intolerant patients with cardiovascular disease or diabetes with end-organ damage | Telmisartan (80 mg) | 80 mg | Placebo | 56 | 80 | 3.73 |
| PROFESS | 20,332 | Recent (<90 days) ischemic stroke | Telmisartan (80 mg) | 80 mg | Placebo | 30 | 70 | 1.75 |
| I-PRESERVE | 4,128 | Heart failure with preserved ejection fraction | Irbesartan (275 mg) | 300 mg | Placebo | 49.5 | 66 | 2.50 |
| ACTIVE-I | 9,016 | Atrial fibrillation plus one risk factor for stroke | Irbesartan (300 mg) | 300 mg | Placebo | 49 | 70 | 2.86 |
| IDNT | 1,715 | Hypertension with diabetic nephropathy | Irbesartan (300 mg) | 300 mg | Placebo | 31 | 76 | 1.96 |
| VAL-HEFT | 5,010 | Heart Failure | Valsartan (254 mg) | 320 mg | Placebo | 23 | 86 | 1.31 |
| VALIANT | 14,703 | Acute Myocardial Infarction | Valsartan (247 mg) or Captopril + Valsartan (116 mg) | 320 mg | Captopril only arm (117 mg), Captopril (107 mg) + Valsartan arm | 24.7 | 83 | 1.32 (Valsartan only), 0.62 (Valsartan + Captopril) |
| VALUE | 15,245 | High-risk hypertensive patients | Valsartan (149 mg) | 320 mg | Amlodipine (8 mg) | 50.4 | 74 | 1.45 |
| NAVIGATOR | 9,306 | Cardiovascular disease with impaired glucose tolerance | Valsartan (153 mg) | 320 mg | Placebo | 60 | 73 | 1.74 |
| CHARM-OVERALL | 7,599 | Heart Failure | Candesartan (24 mg) | 32 mg | Placebo | 37.7 | 79 | 1.86 |
| SCOPE | 4,964 | Elderly with mild-to-moderate hypertension | Candesartan (12 mg) | 32 mg | Placebo | 44.4 | 84 | 1.13 |
| TROPHY | 772 | Prehypertension | Candesartan (16 mg) | 32 mg | Placebo | 24 | NA | 0.77 [c] |
| DIRECT (all) | 5,231 | Diabetes with or without retinopathy | Candesartan (30 mg) | 32 mg | Placebo | 56.8 | 81 | 3.57 |
| LIFE | 9,193 | Hypertension with left ventricular hypertrophy on EKG | Losartan (82 mg) | 100 mg | Atenolol (79 mg) | 57.7 | 82 | 3.23 |

[a] Mean or median dose received by patients according to trial publication

[b] The recommended daily high dose for treatment of hypertension according to package insert of the ARB

[c] Calculated using the average of % compliance of other trials because of missing data

38.9% of patients randomized to ARB) and valsartan (n = 24,455, 33%). A total of 61,197 patients were randomized to control (either placebo or non-placebo control). All of the 15 included trials were double-blind. Baseline patient characteristics of the trials, including background ACE-inhibitor use are presented in the S1 Table.

Funnel plots for publication bias for reporting of cancers and lung cancers with ARB therapy are presented in S1 and S2 Figs, neither of which suggested any publication bias.

## Relationship between cumulative exposure to ARBs and cancer

Meta-regression analysis examining the first co-primary outcome, i.e. the impact of cumulative-exposure to ARBs and risk ratio of all cancers in the ARB arm are presented in Fig 1, Panel A. Overall, there was a statistically significant relationship between cumulative-exposure to ARBs and risk of cancer; with greater degree of cumulative-exposure resulting in a greater risk ratio for cancer in the ARB arm (slope = 0.07 [95% CI 0.03 to 0.11], z = 3.56, p<0.001 with the fixed effect regression method). In the subgroup of patients where there was universal background ACE-inhibitor use in both arms, again there was evidence of a significant relationship between cumulative-exposure to ARBs and risk of cancer; (slope = 0.10 [95% CI 0.03 to 0.18], z = 2.76, p = 0.006 with the fixed effect regression method) (Fig 1, Panel B). Similarly, there was evidence of a significant relationship between cumulative-exposure to ARBs and risk of cancer in patients not receiving ACE-inhibitor treatment in either of the study arms (slope = 0.09 [95% CI 0.03 to 0.16], z = 2.73, p = 0.006 with the fixed effect regression method) (Fig 1, Panel C). Additionally, the relationship between cumulative exposure and risk of cancer with ARBs was statistically significant in both placebo controlled trials (slope = 0.06 [95% CI 0.01 to 0.12], z = 2.27, p = 0.02 with the fixed effect regression method) (Fig 1, Panel D) and in non-placebo controlled trials (slope = 0.09 [95% CI 0.03 to 0.15], z = 2.91, p = 0.004 with the fixed effect regression method) (Fig 1, Panel E).

The effect of randomization to an ARB on occurrence of new cancers was examined according to the degree of cumulative exposure to ARBs. In trials where the average cumulative exposure was > 3 years (two telmisartan trials, one candesartan and one losartan trial), there was a statistically significant excess in new cancers (7.3% vs. 6.2%, $I^2$ = 31.4%, RR 1.11 [95% CI 1.03 to 1.19], p = 0.006 with the fixed effect model, RR 1.12 [95% CI 1.02 to 1.24], p = 0.02 with the random effects model) (Fig 2, Panel A). On the other hand, in lower cumulative-exposure trials (i.e. ≤ 3 years), there was no increased risk of cancer with randomization to an ARB arm (5.5% vs. 6.4%, $I^2$ = 13.7%, RR 0.94 [95% CI 0.89 to 0.99], p = 0.02 with the fixed effect model, RR 0.95 [95% CI 0.89 to 1.00], p = 0.06 with the random effects model) (Fig 2, Panel B).

Meta-analytic risk ratios according to degree of cumulative exposure were calculated in relation to background ACE-inhibitor treatment as well. There was a single trial with cumulative exposure >3 years, where there was ACE-inhibitor treatment in both study arms (i.e. ONTARGET trial, new cancer occurrence 8.4% with ARB+ACE-inhibitor vs. 7.5% ACE-inhibitor only, RR 1.11 [95% CI 1.00 to 1.23], p = 0.05) (Fig 3, Panel A). There was no increase in risk of cancer with ARB+ACE-inhibitor compared to ACE-inhibitor only in trials with cumulative exposure ≤ 3 years) (Fig 3, Panel B). In the trial subsets where there was no ACE-inhibitor treatment in either of the study arms, there was again a statistically significant increase in cancers only if the cumulative exposure was >3 years (Fig 3, Panels C and D).

## Relationship between cumulative exposure to ARBs and lung cancer

Meta-regression analysis examining the second co-primary outcome of cumulative-exposure to ARBs and risk of lung cancer is shown in Fig 4, Panel A. Again, there was a statistically significant relationship between cumulative-exposure to ARBs and risk of lung cancer; with greater degree of cumulative-exposure resulting in a greater risk ratio for lung cancer (slope 0.16 [95% CI 0.05 to 0.27], z = 2.93, p = 0.003 with the fixed effect regression method). There were trends for a positive relationship between cumulative exposure and risk of lung cancer

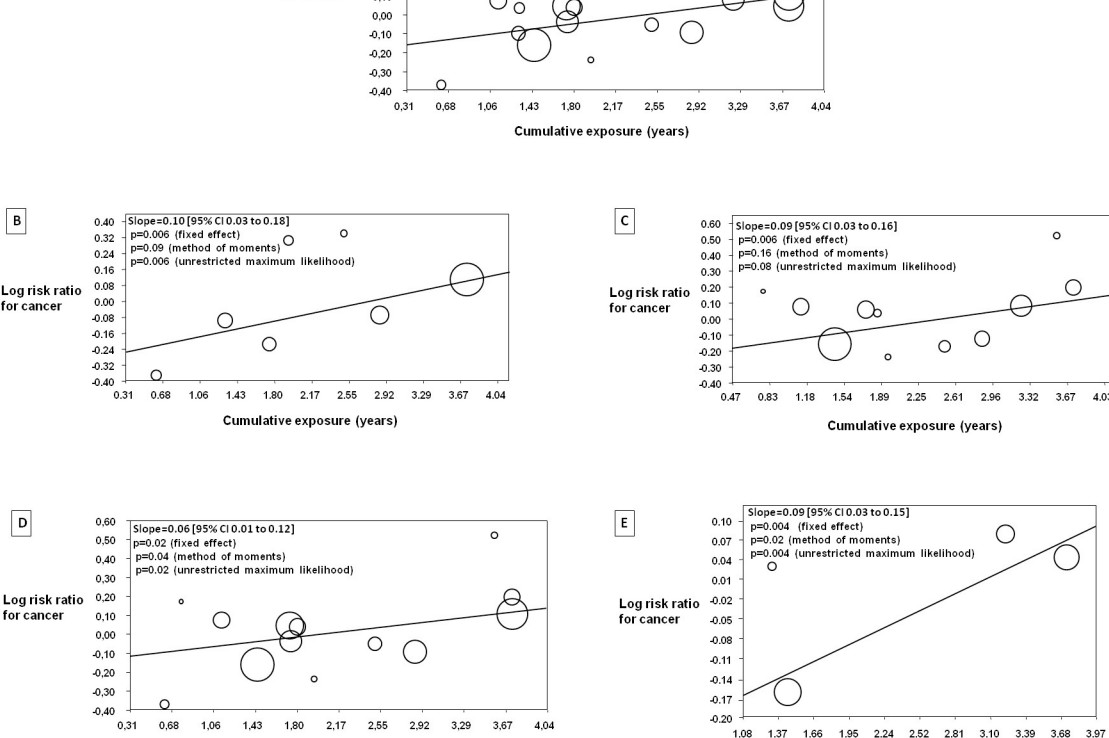

**Fig 1. Meta-regression analysis examining the relationship between cumulative-exposure to ARBs and risk of cancer.** The relationship in all of the included trials is depicted in Panel A, in patients with concomitant background ACE-inhibitor treatment in both arms in Panel B, in patients with no concomitant ACE-inhibitor treatment in either arms in Panel C, in placebo controlled trials in Panel D and in non-placebo controlled trials in Panel E. Each circle represents a clinical trial (or a pairwise comparison in case of trials with 3 treatment groups). The sizes of the circles are proportional to the sample size of trial.

with ARBs in both placebo controlled trials (slope 0.12 [95% CI -0.02 to 0.28], z = 1.68, p = 0.09 with the fixed effect regression method) (Fig 4, Panel B) and in non-placebo controlled trials (slope 0.17 [95% CI -0.009 to 0.36], z = 1.86, p = 0.06 with the fixed effect regression method) (Fig 4, Panel C).

The effect of randomization to an ARB on occurrence of lung cancer was examined according to the degree of cumulative exposure as well. In trials where the average cumulative exposure was > 2.5 years, there was a statistically significant excess in lung cancers (1.2% vs. 0.9%, $I^2$ = 0%, RR 1.21 [95% CI 1.02 to 1.44], p = 0.03 with both the fixed effect model and the random effects model) (Fig 5, Panel A). On the other hand, in lower cumulative-exposure trials, there was no increased risk of lung cancer with ARBs (0.6% vs. 0.8%, $I^2$: 40.9%, RR 0.86 [95% CI 0.73 to 1.01], p = 0.06 with the fixed effect model, RR 0.86 [95% CI 0.69 to 1.09], p = 0.21 with the random effects model) (Fig 5, Panel B).

The relationship between cumulative-exposure to ARBs and risk of prostate and breast cancer were also examined, neither of which showed any significant correlation (p = 0.27 for prostate cancer and, p = 0.71 for breast cancer for all 3 methods).

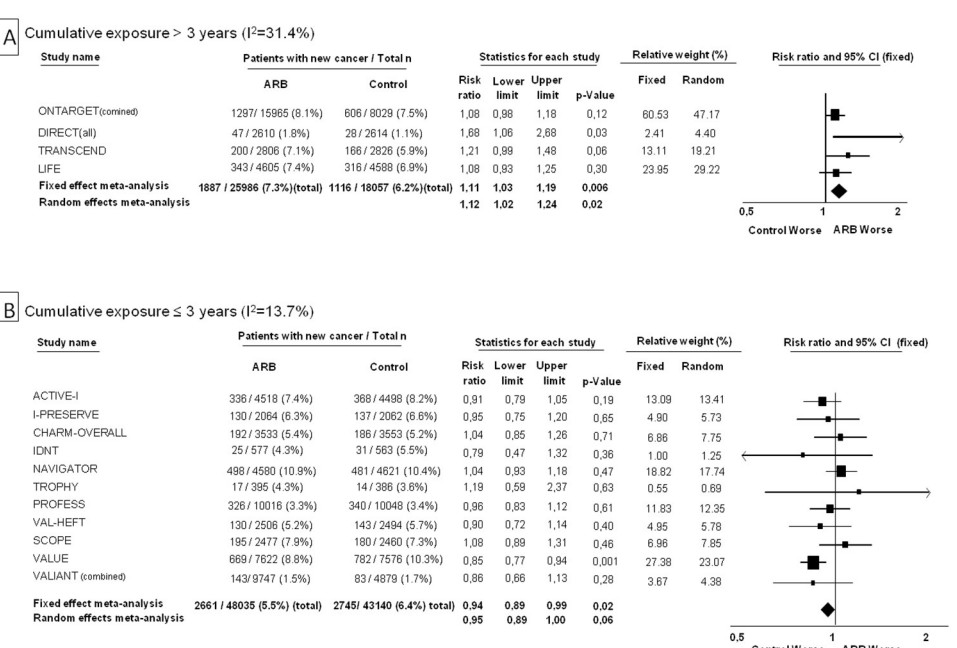

**Fig 2. Forest plots for risk of cancer with ARBs according to level of cumulative exposure.** Panel A shows the plot for cumulative exposure > 3 years (of exposure to daily high dose or equivalent). Panel B shows the plot for cumulative exposure ≤ 3 years.

## Sensitivity analyses and number needed to harm

Sensitivity analyses ruled out the possibility of a single trial being responsible for the relationship between cumulative exposure and risk of cancer and also risk of lung cancer (S2 Table).

Number needed to harm was calculated using current background cancer incidence rates for persons aged 65–69 years corresponding approximately to the mean age of patients enrolled in the trials (i.e. 1610.6 cancers per 100.000 persons per year) [44]. Accordingly, 120 patients needed to be treated with an ARB for 4.7 years (weighted average duration of follow-up of these trials) for one excess cancer diagnosis. The number needed to harm for lung cancer with ARB treatment was 464 patients (for 4.6 years with a background lung cancer incidence rate of 223.1 lung cancers per 100.000 persons per year for persons aged 65–69 years) [44].

## Discussion

This trial-level analysis indicates that risk of cancer increases with increasing cumulative exposure to ARBs. The excess risk of cancer starts to appear after approximately 3 years of exposure to a maximal daily dose of an ARB. The same relationship is also true for lung cancer and this risk becomes statistically significant after 2.5 years of exposure. The excess risk was independent of whether patients received background ACE-inhibitor treatment or not, and whether the trials were placebo or non-placebo controlled.

The first suggestion of a possibly increased cancer risk with ARBs was observed in the CHARM trial in 2003 by Pfeffer et al. [37]. This trial reported an increased risk of fatal cancers in patients randomized to candesartan compared to placebo (p = 0.038). The excess in fatal cancers with the ARB was attributed to "play of chance" by the investigators and was also later reviewed in an FDA document [45]. In 2008, the results of the TRANSCEND and ONTARGET trials, both of which studied telmisartan by Yusuf et al., were published [27, 28]. In the

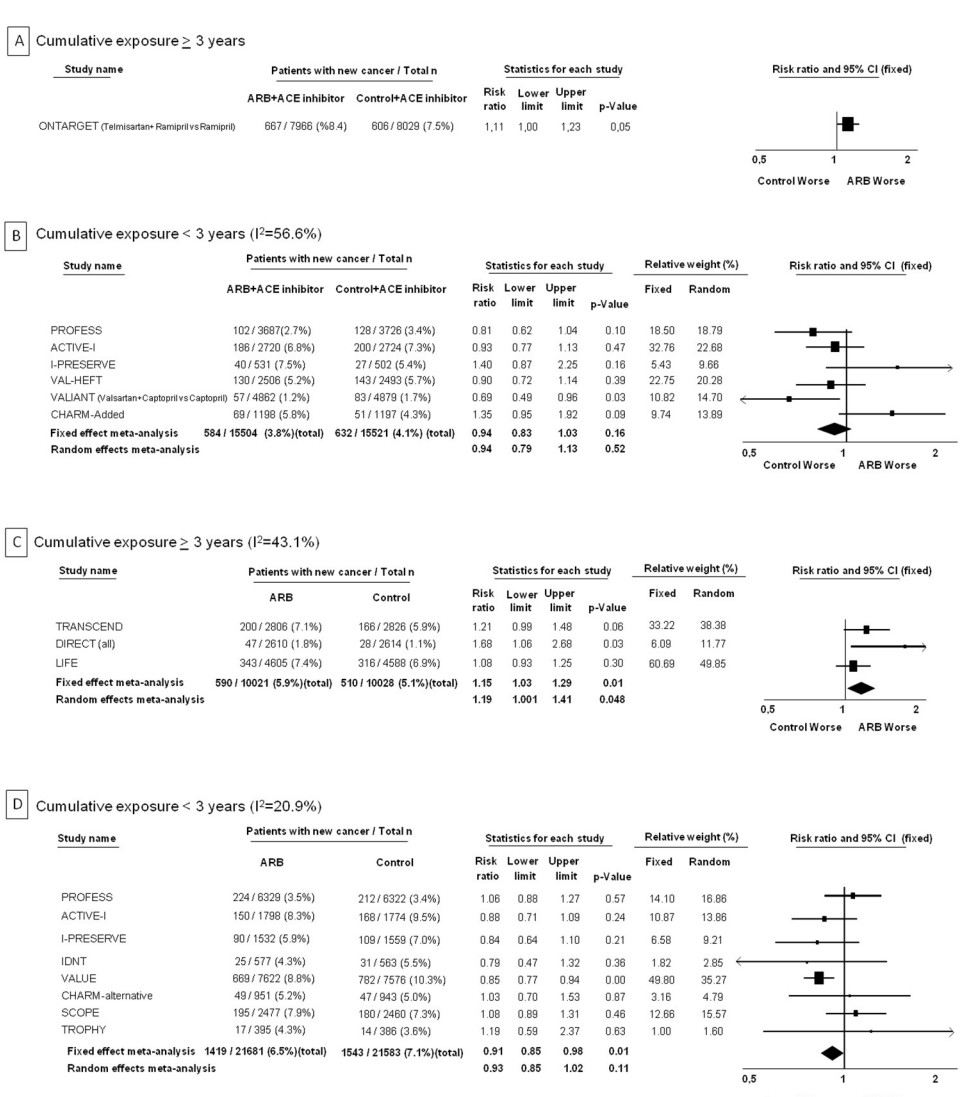

**Fig 3. Forest plots for risk of cancer with ARBs according to level of cumulative exposure and concomitant background ACE-inhibitor treatment.** Panel A shows the plot for cumulative exposure > 3 years (of exposure to daily high dose or equivalent) in patients with concomitant background ACE-inhibitor treatment. Panel B shows the plot for cumulative exposure ≤ 3 years in patients with concomitant background ACE-inhibitor treatment. Panel C shows the plot for cumulative exposure > 3 years in patients without concomitant background ACE-inhibitor treatment. Panel D shows the plot for cumulative exposure ≤ 3 years in patients without concomitant background ACE-inhibitor treatment.

TRANSCEND trial, it was stated "a higher rate of malignancies was observed in patients treated with telmisartan than in those treated with placebo" and "so far, there is no evidence that ARBs are associated with a higher risk of malignancies, chance findings due to multiple testing cannot be excluded" [46]. Similarly, in the related ONTARGET trial it was stated "for malignancies, the hazard ratio of telmisartan+ramipril vs. ramipril only was 1.14 (95% CI 1.03–1.26; p = 0.0089) in all randomised patients and 1.12 (95% CI 1.01, 1.25; p = 0.0366) in patients without cancer at baseline. This finding may be a chance finding due to multiple testing in this trial" [47]. In 2009, a briefing document about telmisartan was also presented to

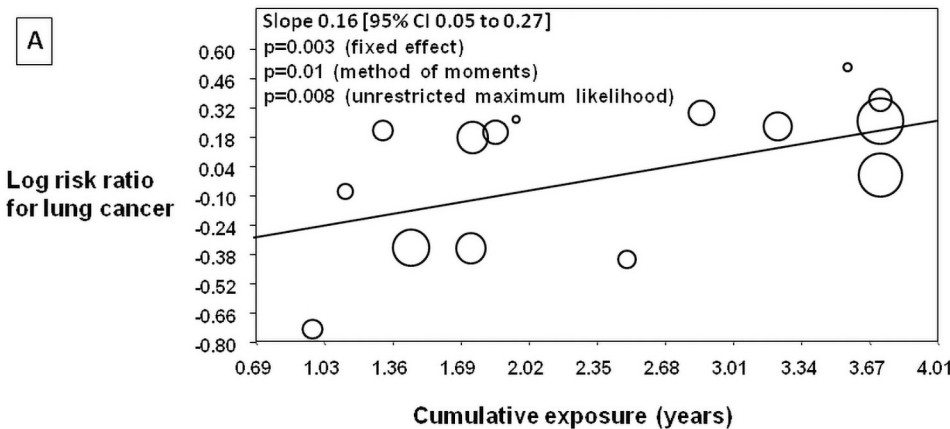

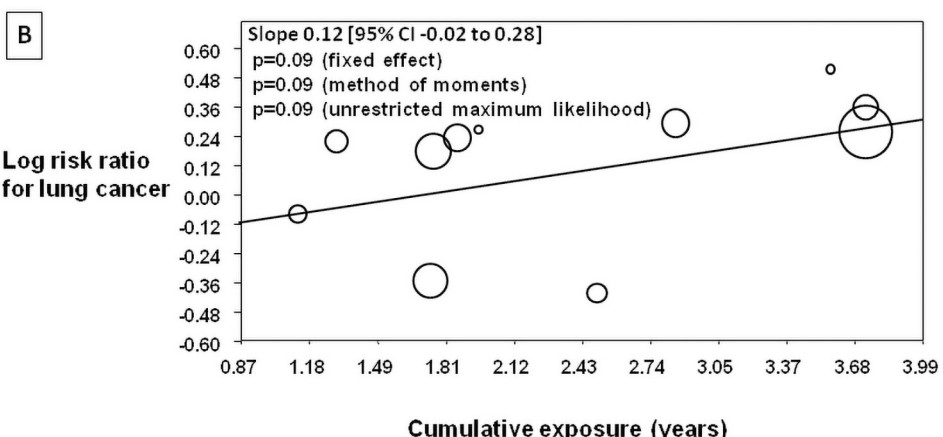

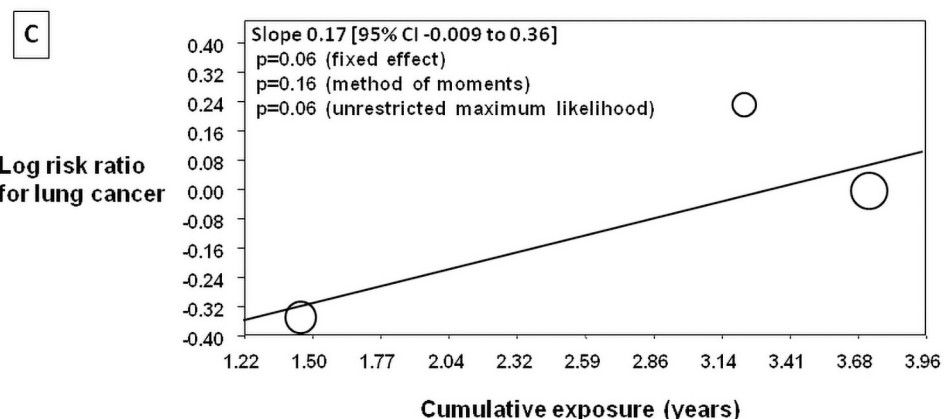

**Fig 4. Meta-regression analysis examining the relationship between cumulative-exposure to ARBs and risk of lung cancer.** The relationship in all of the included trials is depicted in Panel A, in placebo controlled trials in Panel B and in non-placebo controlled trials in Panel C.

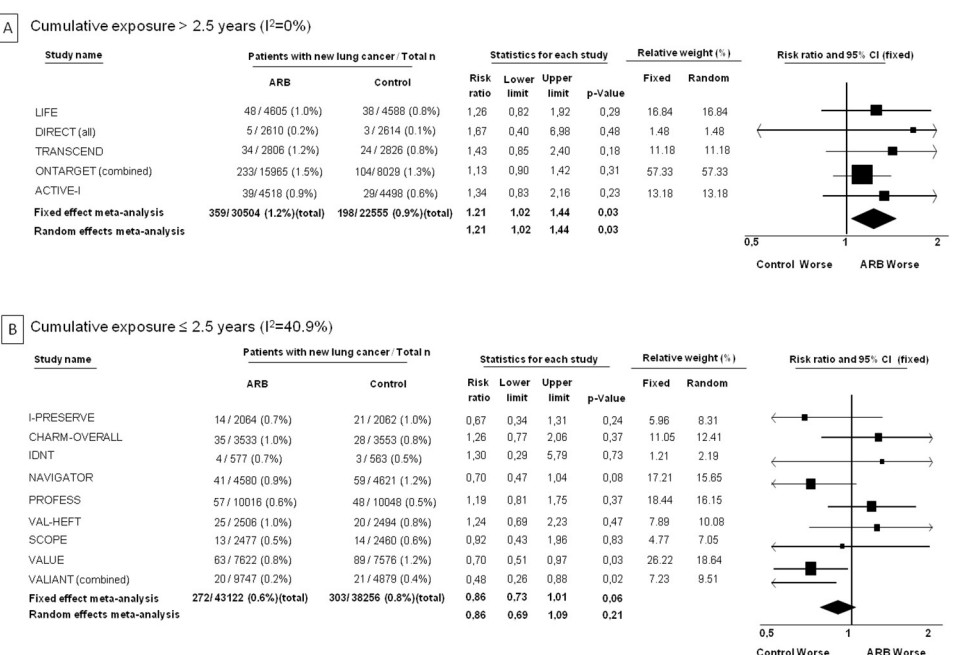

**Fig 5. Forest plots for risk of lung cancer with ARBs according to level of cumulative exposure.** Panel A shows the plot for cumulative exposure > 2.5 years (of exposure to daily high dose or equivalent). Panel B shows the plot for cumulative exposure ≤ 2.5 years.

FDA [4]. This document discussed the above noted excesses in cancers under the heading "Malignancies as a new potential safety signal in patients treated with telmisartan".

Given these findings about ARBs and risk of cancer, we had decided to perform a complete meta-analysis of all publicly available data about cancer occurrence in randomized-controlled trials of ARBs and published our findings in 2010 [14]. Our analysis indicated that patients randomized to ARBs had a significantly increased risk of new cancer occurrence. Among the specific cancers examined, only lung-cancer occurrence was significantly higher in patients randomized to ARBs.

Our 2010 analysis led to calls for additional investigation of the cancer risk with ARBs to be performed by regulatory agencies [48]. Consequently, the FDA performed a trial level meta-analysis of preexisting trials. FDA's analysis included 155,816 patients from 31 randomized trials and concluded that there is no increased risk of cancer with ARBs [15]. However, the FDA did not publish any details of their methods or results, such as the names of the trials, rates of cancers in the exposed and unexposed patients in each trial or the degree of cumulative exposure in any of the trials. Individual patient-level data analysis enabling robust time to event calculations and examination of cumulative exposure-risk relationship were also not done, which would be critical while examining the risk of a slowly developing adverse event such as cancer. FDA's analysis was later scrutinized by a senior FDA team leader, Dr. Thomas A. Marciniak [49, 50]. For example, according to Marciniak's investigation, adverse events reported as "lung carcinomas" were not considered as lung cancer in FDA's analysis. Because of the limitations in the agency's official investigation, this FDA scientist performed an individual patient-level analysis with the data submitted to the FDA. Marciniak's analysis used data from a total of 11 trials, including the ONTARGET, TRANSCEND, LIFE, CHARM, PROFESS, IDNT, VAL-HEFT and VALIANT trials, which were included in the current study as well. Marciniak's patient-level analysis identified a 24% increase in the risk of lung cancer with ARBs

(p = 0.003), a risk increase comparable to the 21% risk increase in lung cancers calculated in the current analysis. Marciniak's analysis was posted in detail at the FDA website in 2015 [51]. Kaplan-Meier analyses of this data estimated approximately 0.8 excess lung cancer cases per year per 1,000 patients treated. Additionally, according to this analysis, significantly more patients randomized to ARBs died with lung cancer (HR = 1.27, 95% CI 1.08–1.51, P = 0.005). Given these findings, it was recommended that "the increased risk of lung cancer with all ARBs should be described in labeling" [51]. Additionally, in Marciniak's individual patient-level analysis, the shapes of the incidence curves for lung cancer were considered to be consistent with a cancer promoter effect of ARBs; there was a delayed initial divergence of the rates in ARB and control arms followed by continuing divergence throughout the duration of follow-up.

In addition to the investigations performed by the regulatory agencies, several other analyses examining risk of cancer with ARBs were also published after our 2010 meta-analysis [17–26]. The conclusions of these analyses were highly conflicting, some suggesting no excess risk [21, 24, 26] and others suggesting an increased cancer risk [22, 23]. Importantly, the relationship between exposure and risk was again not assessed in any of the reported analyses of the randomized trials. The current study shows that risk of cancer with ARBs increases with increasing exposure at a trial-level. This relationship at least partially explains the heterogeneity in the results of the investigations examining the ARB-cancer issue in randomized trials. Accordingly, if the analysis mainly includes long-term, high exposure trials, there is a significant increase in overall cancers and lung cancer [14]. On the other hand, if an analysis includes a high number of patients with low exposure to ARBs, the excess risk of cancer coming from high exposure trials is diluted. This causes a bias towards the null [25]. While this bias could be overcome by examining the risk according to exposure of each patient, such analyses were unfortunately not performed by the regulatory agencies or by the ARB Trialists Collaboration.

One previous meta-analysis had suggested that the excess risk of cancer with ARBs may be limited to patients with concurrent ACE-inhibitor treatment [24]. Our results indicate that the cumulative exposure-risk relationship exists regardless of whether patients receive concurrent ACE-inhibitor treatment. Moreover, there is a statistically significant increase in the risk of cancer with ARBs even in patients without background ACE-inhibitor treatment, as long as there is enough exposure. Similarly, the results were not different according to the control type (i.e. placebo control or active control).

In 2018, regulatory agencies including the FDA, identified NDMA, a possible human carcinogen in several formulations of valsartan, a commonly used ARB [6]. Subsequently, recalls of these drug products were ordered across the globe. This recall was expanded several times to include more valsartan containing drug products, because NDMA was identified in them as well [7]. Consequently, the FDA announced that they have started testing all the other drugs in the ARB class for NDMA [8]. In this announcement, FDA commissioner Scott Gottlieb stated that the synthesis of other ARBs can have similarities to the synthesis of valsartan, and this genotoxic molecule can be a common impurity that develops during synthesis of all ARBs. The FDA added that their tests will continue until they identify all products that may contain NDMA in the ARB class, and they are no longer available in the United States. In the following months, NDEA (a similar, again possibly carcinogenic nitrosamine) was identified in several valsartan, losartan and irbesartan containing drug products originating from different manufacturers of the active pharmaceutical ingredient, again resulting in recalls [9–11]. Subsequently in 2019, a third nitrosamine, N-nitroso-N-methyl-4-aminobutyric acid (NMBA), a known animal and potential human carcinogen, was found in several losartan formulations, which resulted in additional recalls [12]. Moreover, throughout most of 2021 multiple lots of several ARBs including irbesartan, losartan and valsartan were recalled, this time due to

another potentially carcinogenic impurity, namely azido compounds [13]. Testing of many other ARB containing drug products are still underway and how much of the ARB drug class will ultimately be affected by possibly carcinogenic impurities is not yet known. It is also unknown whether the specific formulations of ARB drug products used in the trials included in the current analysis contained nitrosamines or azido compounds. Therefore, whether an impurity developing during synthesis of ARBs is the mechanism for the increase in new cancers with ARBs is unclear. Alternatively, previous studies using mouse models and cancer cell lines have directly implicated the renin-angiotensin system in the regulation of cell proliferation, angiogenesis, tumor expansion, as well as metastasis [52, 53]. For example, evidence indicates that angiotensin II receptor type-1 (AT1R) blockade with an ARB, which results in unopposed angiotensin II receptor type-2 (AT2R) stimulation is capable of causing tumor angiogenesis in vivo [54]. Therefore, the exact mechanism of the increased cancer risk with ARBs is currently not clear. On the other hand, it should be noted that while the allied class of ACE-inhibitors was associated with an increase in lung cancer in one retrospective cohort study with possible residual confounding [55], ACE-inhibitors had no effect on incident cancer in long-term randomized controlled trials including more than 60.000 patients (RR 1.01 [95% CI 0.95 to 1.07]) [56].

The FDA recently estimated that if 8,000 people took the highest valsartan dose (320 mg) from NDMA-affected medicines daily for 4 years, there may be one additional case of cancer over the lifetimes of these 8,000 people [7]. However, number needed to harm according to the current analysis is remarkably lower; 120 patients needed to be treated with the maximal daily dose of an ARB for 4.7 years for one excess cancer diagnosis. In 2011, it was calculated that about 200 million individuals are treated with an ARB globally [5]. Given the numbers needed to harm of 120 for one excess cancer and 464 for one excess lung cancer, it can be projected that if 200 million patients are exposed to daily high doses for 4.7 years (or equivalent), approximately 1.7 million excess cancers (and 430.000 lung cancers in 4.6 years) could be potentially caused by this class of drugs. On the other hand, if ARBs had been superior to other classes of drugs in terms of blood pressure reduction or prevention of cardiovascular events, such benefits could potentially offset the excess cancer risk associated with them. However, there is actually evidence that ARBs may be inferior to many other classes of antihypertensives for prevention of mortality and cardiac morbidity. For example, while ACE inhibitors reduce total mortality and risk of myocardial infarction in hypertensives, ARBs do not reduce the risk of either of these outcomes [57–61]. ARBs have actually never been shown to reduce myocardial infarctions, even in placebo controlled trials [3, 29, 36, 58, 62]. Likewise, an ARB, namely valsartan, has been shown to be significantly inferior to an active control (i.e. amlodipine, a calcium channel blocker) for prevention of myocardial infarctions [35]. On the other hand, there is no evidence that other antihypertensive medications contain carcinogenic impurities or raise the risk of cancers in randomized trials [56, 63]. Therefore, other classes of antihypertensives with good safety and efficacy data (such as ACE-inhibitors, calcium-channel blockers or others) should become the preferred first-line agents in the treatment of hypertension.

## Limitations

The fact that this is a trial level analysis is a limitation of this study. However, individual patient level data was not publicly available. Nevertheless, the study included 15 different clinical trials with enough variation in both cumulative exposure and the outcome measures, enabling meaningful meta-regression analysis. On the other hand, the impact of gender, age and smoking on the current findings could still not be examined due to lack of patient level data. However, the current analysis included only randomized controlled trials, therefore the likelihood

of confounding variables being responsible for the increased cancer risk with ARBs in high-exposure trials is unlikely. It should also be remembered that ARBs did not prolong survival in these high-exposure trials [27, 28, 40, 41]. Therefore, it is also unlikely that competing outcomes (i.e. death vs. cancer) are responsible for the observed findings. However, more robust time to event analysis could still not be performed due to lack of individual patient level data. It is possible that excess risk of cancer in the later years of ARB treatment can be much greater than the earlier years because of the latency period. On the other hand, Marciniak used individual patient-level data of most of the trials included in the current trial-level analysis and noted that shapes of the incidence curves for lung cancer were considered to be consistent with a cancer promoter effect of ARBs [51]. He observed a delayed initial divergence of cancer rates in ARB and control arms, which corroborates our finding of an increased cancer risk only with increasing cumulative exposure. In this context, it should be noted that cumulative exposure is the product of duration of exposure and dose. Since duration of exposure and cumulative-exposure closely correlate by definition, it is not possible to determine whether the relationships observed in the current analysis merely reflect the impact of duration of exposure rather than the impact of cumulative exposure. It should also be noted that the current analysis is not able to determine whether the mechanism of increased cancer risk with ARBs is related to the carcinogenic impurities recently identified in several ARB containing drug products.

## Conclusions

This analysis shows that risk of cancer and specifically lung cancer increase with increasing cumulative exposure to ARBs. The relationship between cumulative exposure to ARBs and cancer risk explains the heterogeneity in the results of randomized trials, since trials were highly heterogeneous in terms of cumulative exposure. Detailed and impartial analysis of the vast amount of patient-level data of randomized trials that the regulatory agencies already have, including examination of cumulative exposure—risk relationship, can confirm the current findings. Because of the ongoing widespread use of ARBs globally, their potential of excess cancer risk with long-term use has profound implications for patients and prescribing clinicians.

## Supporting information

**S1 Fig. Funnel plot for examining publication bias according reporting of cancers.**
(DOCX)

**S2 Fig. Funnel plot for examining publication bias according reporting of lung cancer.**
(DOCX)

**S1 Table. Patient characteristics of the randomized controlled trials included in the analysis.**
(DOCX)

**S2 Table. Sensitivity analyses with the one-study out method.**
(DOCX)

## Author Contributions

**Conceptualization:** Ilke Sipahi.

**Data curation:** Ilke Sipahi.

**Formal analysis:** Ilke Sipahi.

**Investigation:** Ilke Sipahi.

**Methodology:** Ilke Sipahi.

**Project administration:** Ilke Sipahi.

**Resources:** Ilke Sipahi.

**Software:** Ilke Sipahi.

**Supervision:** Ilke Sipahi.

**Validation:** Ilke Sipahi.

**Writing – original draft:** Ilke Sipahi.

**Writing – review & editing:** Ilke Sipahi.

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
