## [Decision Letter · Decision Letter 0]

20 Dec 2021

PONE-D-21-28723Relationship between cumulative-exposure to angiotensin-receptor blockers and risk of cancer in randomized controlled trialsPLOS ONE

Dear Dr. Ilke Sipahi,

Thank you for submitting your manuscript to PLOS ONE. After careful consideration, we feel that it has merit but does not fully meet PLOS ONE’s publication criteria as it currently stands. Therefore, we invite you to submit a revised version of the manuscript that addresses the points raised during the review process.

Please respond to each of the points made below by myself and Reviewer 1.  Most importantly respond to the discrepancies in the numbers and double counting of the control group pointed out by Reviewer 1.  

We look forward to receiving your revised manuscript.

Kind regards,

James M Wright

Academic Editor

PLOS ONE

“I have read the journal's policy and the authors of this manuscript have the following competing interests:

Dr. Sipahi has received lecture honoraria from Novartis, Boehringer-Ingelheim, Sanofi, Sandoz, Bristol-Myers Squibb, Bayer, Pfizer, Ranbaxy, Servier and ARIS and served on advisory board for Novartis, Sanofi, Servier, Bristol-Myers Squibb, Pfizer, Bayer and I.E. Ulagay.”

Additional Editor Comments:

I think you have underplayed the implications of your findings. This has profound implications for the long-term treatment of hypertension, because of the current widespread use of ARBs and the availability of safer alternatives. To emphasize the importance of your findings.

1) Have the title state your conclusions.

2) In the discussion estimate the numbers of excess cancers and lung cancers being caused by these drugs (my guess is that it is in the 10's of 1000's.

3) Discuss the safer alternatives in the discussion.

4) Include in your conclusions that this has profound implications for patients and clinicians.

5) Include in the conclusions how these critical findings can be confirmed with future research.

Reviewers' comments:

Reviewer's Responses to Questions

**Comments to the Author**

1. Is the manuscript technically sound, and do the data support the conclusions?

Reviewer #1: Partly

Reviewer #2: Yes

2. Has the statistical analysis been performed appropriately and rigorously? 

Reviewer #1: No

Reviewer #2: Yes

3. Have the authors made all data underlying the findings in their manuscript fully available?

Reviewer #1: Yes

Reviewer #2: Yes

4. Is the manuscript presented in an intelligible fashion and written in standard English?

Reviewer #1: Yes

Reviewer #2: Yes

5. Review Comments to the Author

Reviewer #1: This research article is a clinically important review article examining the relationship of cumulative exposure to ARBs and the increased risk of all new cancers as well as specific increase in lung cancer and is important review article that needs to be published.

The authors claimed to have used publicly available data from the ARB Trialist Collaboration 2011. However, Zhao et al 2016 meta-analysis of cancer risk with ARBs article, a more recent publication is missing in the reference section (see details of the reference below) and numerator and denominators reported of various studies included in this review do not match the Zhao et al 2016 article.

Zhao Y-T, Yang –Peng Li, Zhang J-Q, Wang L and Zhong Yi. Angiotensin II receptor blockers and cancer risk. A meta-analysis of randomized controlled trials. Medicine. May 2016; Volume 95 (18): 1-7.

Figure 2 data discrepancies in this review article

1. Overall cancer numbers in Zhao et al 2016 article has 3 values for CHARM studies -CHARM ALTERNATIVE 2003 – ARB + ACEI = 49/951 vs ACEI + PbO = 47/943; CHARM ADDED 2003- ARB + ACEI = 69/1198 vs ACEI = 51/1197; and CHARM OVERALL 2003 – ARB =104/3803 vs ACEI = 111/3796

However, CHARM OVERALL in this article (Sipahi et al 2021) is reported as - 192/3533 vs 186/353 (??)

2. DIRECT study – Denominators differ

Zhao et al 2016 - ARB = 47/2613 vs control 28/=2618

Sipahi et al 2021 - ARB = 47/2610 vs control 28/= 2614

3. TRANSCEND study numerator and denominators differ

Zhao et al 2016 – ARB = 236/2954 vs control = 204/2972;

Sipahi et al 2021 : ARB = 200/2806 vs 166/2826 )

4. IDNT study denominator differ

Zhao et al 2016 – ARB = 25/ 579 vs 31/569

Sipahi et al 2021. 25/577 vs 31/563

5. ONTARGET study denominators differ

Zhao et al 2016 –ARB = 630/8542 vs ACEI= 606/8576

Sipahi et al 2021 – ARB = 630/7999 vs ACEI = 606/8029

Also please note that the denominators used in Figure 5 for lung cancer meta-analysis in Sipahi et al 2021 ARB = 100/8542 and control = 101/8576 differ from the denominators used in Fig 2 in their own article and are similar to Zhao et al 2016 numbers reported above.

Zhao et al 2016 – (ARB + ACEI)= 667/ 8502 and ACEI = 606/8576

Sipahi et al 2021 used – (ARB + ACEI)= 667/ 7966 and ACEI = 606/8029

6. VALIANT study denominator differ

Zhao et al 2016 ARB = 86/4909 vs ACEI = 83/4909

Sipahi et al 2021 ARB = 86/4885 vs ACEI = 83/4879

Double counting of control groups in the meta-analysis of this article Sipahi et al

Figure 2A meta analysis of all cancer risk used ONTARGET control group twice (N = 8029) in the meta-analysis leading to over counting of including patients in the overall RR and gives more weight to the study.

Similarly in Figure 2B, VALIANT study control group was used twice (N = 4879). This double counts the same participants in the control group and does not provide an accurate estimate of relative risk of cancer.

The authors are recommended to check accuracy of their data input and avoid double counting the control groups ONTARGET and VALIANT studies to calculate RR of all cancers as well as lung cancer.

A well written review article and a clinically important topic that should be published after the above mentioned errors are corrected.

Reviewer #2: This study has led to compelling and likely impactful findings that will further fuel the controversy pertaining to the link between ARBs and cancer risk. Demonstrating an increased risk of overall cancer and lung cancer with ARBs is related to the degree of cumulative exposure may explain the contradictory findings of previous studies; however, the fact that this is a trial-level analysis that includes data for only half of the patients included in the FDA and ARB Trialists Collaboration patient-level analyses is an important limitation that cannot be overcome without full data availability from all RCTs. I am hoping the publication of the author’s study will prompt a more definitive analysis of all individual patient data by an independent group of researchers with no conflicts, or release of the full analysis already conducted by the FDA and/or ARB Trialists Collaboration.

6. PLOS authors have the option to publish the peer review history of their article (what does this mean?). If published, this will include your full peer review and any attached files.

Reviewer #1: No

Reviewer #2: No

---

## [Author Response · Author response to Decision Letter 0]

6 Jan 2022

Response to Reviewers for PONE-D-21-28723 title'Risk of cancer with angiotensin-receptor blockers increases with increasing cumulative-exposure: Meta-regression analysis of randomized trials''

Reviewer #1

Reviewer Comment: This research article is a clinically important review article examining the relationship of cumulative exposure to ARBs and the increased risk of all new cancers as well as specific increase in lung cancer and is important review article that needs to be published.

The authors claimed to have used publicly available data from the ARB Trialist Collaboration 2011. However, Zhao et al 2016 meta-analysis of cancer risk with ARBs article, a more recent publication is missing in the reference section (see details of the reference below) and numerator and denominators reported of various studies included in this review do not match the Zhao et al 2016 article.

Zhao Y-T, Yang –Peng Li, Zhang J-Q, Wang L and Zhong Yi. Angiotensin II receptor blockers and cancer risk. A meta-analysis of randomized controlled trials. Medicine. May 2016; Volume 95 (18): 1-7.

Response: I thank reviewer #1 for the thorough and very helpful remarks. I revised the paper extensively according these comments. First, I regret that I did not include the above mentioned manuscript by Zhao Y-T and colleagues. I have now included this relevant article in the revised manuscript (reference 26).

-

Reviewer Comment: Figure 2 data discrepancies in this review article

1. Overall cancer numbers in Zhao et al 2016 article has 3 values for CHARM studies -CHARM ALTERNATIVE 2003 – ARB + ACEI = 49/951 vs ACEI + PbO = 47/943; CHARM ADDED 2003- ARB + ACEI = 69/1198 vs ACEI = 51/1197; and CHARM OVERALL 2003 – ARB =104/3803 vs ACEI = 111/3796

However, CHARM OVERALL in this article (Sipahi et al 2021) is reported as - 192/3533 vs 186/353 (??)

Response: I thank the reviewer for this comment. The CHARM-OVERALL trial had three components. CHARM-ALTERNATIVE, CHARM-ADDED and CHARM-PRESERVED. Data from these 3 components were added to form the CHARM-OVERALL data. It is apparent that the data in the Zhao et al. article for CHARM-OVERALL is incomplete. For example, Zhao et al. report 49 cancers for the ARB arm in CHARM-ALTERNATIVE, and 69 cancers for the same arm of CHARM-ADDED. These 2 components combined yields a total cancer number of 118, without the addition of cancers from the 3rd component of CHARM-PRESERVED. Yet, Zhao et al. report a total of only 104 cancers in the ARB arm of CHARM-OVERALL. However, data provided from the industry to the ARB Trialists Collaboration report 192 cancers in the ARB arm and 186 cancers in the placebo arm of CHARM-OVERALL trial, which was also the source of data for the current analysis. 

-

Reviewer Comment: 2. DIRECT study – Denominators differ

Zhao et al 2016 - ARB = 47/2613 vs control 28/=2618

Sipahi et al 2021 - ARB = 47/2610 vs control 28/=2614

Response: In the current analysis, we used data from the ARB Trialists Collaboration which was dedicated to examining the risk of cancers with data provided from the sponsors of these trials, specifically for this purpose. Zhao et al. obtained their data from original trial publications, which were not dedicated to cancer outcomes. The ARB Trialists Collaboration typically reports smaller numbers for denominators compared to original trial publications (dedicated to primary outcomes) since data for cancer outcomes were missing in some of the patients and these patients could not be included in the denominator . 

-

Reviewer Comment:

3. TRANSCEND study numerator and denominators differ

Zhao et al 2016 – ARB = 236/2954 vs control = 204/2972;

Sipahi et al 2021 : ARB = 200/2806 vs 166/2826 )

Response: Again, we used data from the ARB Trialists Collaboration which states ''included were patients who were cancer free at baseline'' in Fig 1 of their publication (J Hypertens 2011;29:623-635), hence the smaller numbers for the numerator and denominators for both arms in TRANSCEND study for the current analysis. 

-

Reviewer Comment:

4. IDNT study denominator differ

Zhao et al 2016 – ARB = 25/ 579 vs 31/569

Sipahi et al 2021. 25/577 vs 31/563 5. ONTARGET study denominators differ

Zhao et al 2016 –ARB = 630/8542 vs ACEI= 606/8576

Sipahi et al 2021 – ARB = 630/7999 vs ACEI = 606/8029

Also please note that the denominators used in Figure 5 for lung cancer meta-analysis in Sipahi et al 2021 ARB = 100/8542 and control = 101/8576 differ from the denominators used in Fig 2 in their own article and are similar to Zhao et al 2016 numbers reported above.

Zhao et al 2016 – (ARB + ACEI)= 667/ 8502 and ACEI = 606/8576

Sipahi et al 2021 used – (ARB + ACEI)= 667/ 7966 and ACEI = 606/8029

6. VALIANT study denominator differ Zhao et al 2016 ARB = 86/4909 vs ACEI = 83/4909

Sipahi et al 2021 ARB = 86/4885 vs ACEI = 83/4879

Response: Again, the source of data for the current analysis typically reports smaller numbers for denominators compared to original trial publications (and therefore compared to Zhao et al) because data for cancer outcomes were missing in some of the patients and these patients could not be included in the denominator. 

-

Reviewer Comment: Double counting of control groups in the meta-analysis of this article Sipahi et al Figure 2A meta analysis of all cancer risk used ONTARGET control group twice (N = 8029) in the meta-analysis leading to over counting of including patients in the overall RR and gives more weight to the study.

Similarly in Figure 2B, VALIANT study control group was used twice (N = 4879). This double counts the same participants in the control group and does not provide an accurate estimate of relative risk of cancer.

The authors are recommended to check accuracy of their data input and avoid double counting the control groups ONTARGET and VALIANT studies to calculate RR of all cancers as well as lung cancer.

A well written review article and a clinically important topic that should be published after the above mentioned errors are corrected.

Response: I thank the reviewer for this warning. Figures 2A and 2B (for total cancers) and Figure 5A and 5B (for lung cancer) have been updated as requested by the reviewer. The relevant sections of the text have been updated as well.

Reviewer #2

Reviewer Comment: This study has led to compelling and likely impactful findings that will further fuel the controversy pertaining to the link between ARBs and cancer risk. Demonstrating an increased risk of overall cancer and lung cancer with ARBs is related to the degree of cumulative exposure may explain the contradictory findings of previous studies; however, the fact that this is a trial-level analysis that includes data for only half of the patients included in the FDA and ARB Trialists Collaboration patient-level analyses is an important limitation that cannot be overcome without full data availability from all RCTs. I am hoping the publication of the author’s study will prompt a more definitive analysis of all individual patient data by an independent group of researchers with no conflicts, or release of the full analysis already conducted by the FDA and/or ARB Trialists Collaboration.

Response: I thank the reviewer for the comments. I added the following sentence to the conclusions section:

‘‘Detailed and impartial analysis of the vast amount of patient-level data of randomized trials that the regulatory agencies already have, including examination of cumulative exposure - risk relationship, can confirm the current findings.’’

Additional Editor Comments

I think you have underplayed the implications of your findings. This has profound implications for the long-term treatment of hypertension, because of the current widespread use of ARBs and the availability of safer alternatives. To emphasize the importance of your findings.

Reviewer Comment: 1) Have the title state your conclusions.

Response: I changed the title as follows: ''Risk of cancer with angiotensin-receptor blockers increases with increasing cumulative-exposure: Meta-regression analysis of randomized trials''.

-

Reviewer Comment: 2) In the discussion estimate the numbers of excess cancers and lung cancers being caused by these drugs (my guess is that it is in the 10's of 1000's.)

Response: In their article published in 2011 Volpe et al stated ''it is calculated that about 200 million individuals are treated with ARBs on our planet''. Given the numbers needed to harm of 120 for one excess cancer and 464 for one excess lung cancer, it can be projected that if 200 million patients are exposed to daily high doses for 4.7 years (or equivalent), approximately 1.7 million excess cancers (and 430.000 lung cancers in 4.6 years) could be potentially caused by this class of drugs. We added all of these calculations to the discussion section.

-

Reviewer Comment: 3) Discuss the safer alternatives in the discussion.

Response: I added the following to the discussion section: 

‘‘On the other hand, if ARBs had been superior to other classes of drugs in terms of blood pressure reduction or prevention of cardiovascular events, such benefits could potentially offset the excess cancer risk associated with them. However, there is actually evidence that ARBs may be inferior to many other classes of antihypertensives for prevention of mortality and cardiac morbidity. For example, while ACE inhibitors reduce total mortality and risk of myocardial infarction in hypertensives, ARBs do not reduce the risk of either of these outcomes [57-61]. ARBs have actually never been shown to reduce myocardial infarctions, even in placebo controlled trials [3,29,36,58,62,] . Likewise, an ARB, namely valsartan, has been shown to be significantly inferior to an active control (i.e. amlodipine, a calcium channel blocker) for prevention of myocardial infarctions [35]. On the other hand, there is no evidence that other antihypertensive medications contain carcinogenic impurities or raise the risk of cancers in randomized trials [56,63]. Therefore, other classes of antihypertensives with good safety and efficacy data (such as ACE-inhibitors, calcium-channel blockers or others) should become the preferred first-line agents in the treatment of hypertension.''

-

Reviewer Comment: 4) Include in your conclusions that this has profound implications for patients and clinicians.

Response: I changed the last sentence of the conclusions section as follows:

‘‘Because of the ongoing widespread use of ARBs globally, their potential of excess cancer risk with long-term use has profound implications for patients and prescribing clinicians. ’’

-

Reviewer Comment: 5) Include in the conclusions how these critical findings can be confirmed with future research.

Response:I added the following sentence to the conclusions section:

‘‘Detailed and impartial analysis of the vast amount of patient-level data of randomized trials that the regulatory agencies already have, including examination of cumulative exposure risk relationship, can confirm the current findings.’’

---

## [Editor Report · Decision Letter 1]

20 Jan 2022

Risk of cancer with angiotensin-receptor blockers increases with increasing cumulative-exposure: Meta-regression analysis of randomized trials''

PONE-D-21-28723R1

Dear Dr. Sipahi,

We’re pleased to inform you that your manuscript has been judged scientifically suitable for publication and will be formally accepted for publication once it meets all outstanding technical requirements.

Kind regards,

James M Wright

Academic Editor

PLOS ONE
---

## [Editor Report · Acceptance letter]

4 Feb 2022

PONE-D-21-28723R1 

Risk of cancer with angiotensin-receptor blockers increases with increasing cumulative exposure: Meta-regression analysis of randomized trials 

Dear Dr. Sipahi:

I'm pleased to inform you that your manuscript has been deemed suitable for publication in PLOS ONE. Congratulations! Your manuscript is now with our production department. 

Kind regards, 

on behalf of

Professor James M Wright 

Academic Editor

PLOS ONE